# An Interference-Free Voltammetric Method for the Detection of Sulfur Dioxide in Wine Based on a Boron-Doped Diamond Electrode and Reaction Electrochemistry

**DOI:** 10.3390/ijms241612875

**Published:** 2023-08-17

**Authors:** Eva Culková, Zuzana Lukáčová-Chomisteková, Renata Bellová, Miroslav Rievaj, Jarmila Švancarová-Laštincová, Peter Tomčík

**Affiliations:** 1Electroanalytical Chemistry Laboratory, Department of Chemistry and Physics, Faculty of Education, Catholic University in Ružomberok, Hrabovská Cesta 1, SK-034 01 Ružomberok, Slovakia; eva.culkova@ku.sk (E.C.); zuzana.lukacova@gmail.com (Z.L.-C.); renata.bellova@ku.sk (R.B.); miroslav.rievaj@ku.sk (M.R.); 2Central Controlling & Testing Institute for Agriculture in Bratislava, Matúškova 21, SK-833 16 Bratislava, Slovakia

**Keywords:** voltammetry, sulfur dioxide, wine, iodide, iodine, boron-doped diamond

## Abstract

This paper describes a new, simple, and highly selective analytical technique for the detection of sulfur dioxide in wine, as a real sample with a relatively complicated matrix. The detection of the above analyte was based on the electrogeneration of iodine from iodide on a boron-doped diamond electrode, without modifications, in the presence of 0.1 mol dm^−3^ HClO_4_ as a supporting electrolyte. The electrogenerated iodine reacted with sulfur dioxide, forming iodide ions and sulfuric acid (i.e., a Bunsen reaction). The product of this reaction, the iodide ion, diffused back to the surface of the boron-doped diamond electrode and oxidized itself again. This chemical redox cycling enhanced the voltammetric response of the boron-doped diamond electrode. The selectivity of the determination was assured using NaOH and formaldehyde during sample preparation, and a blank was also measured and taken into account. The detection limit was estimated to be 10^−6^–10^−7^ mol dm^−3^. However, the content of sulfur dioxide in wine is significantly higher, which can lead to more accurate and reliable results.

## 1. Introduction

Sulfur dioxide is a colorless, toxic, inorganic gas that has no odor at low concentrations; however, it has a sharp and unpleasant odor at very high concentrations. It has an irritating effect on the mucous membranes of the human respiratory tract and eyes. In nature, it prevents the photosynthesis of plants. It has whitening effects, and dissolves easily in water and wine due to the formation of sulfurous acid H_2_SO_3_ [1]. From a wine-technology point of view, sulfur dioxide has a wide spectrum of positive effects on wine, and still has an irreplaceable position, because there is no suitable compound that has similar effects on wine. Due to the simplicity of applying SO_2_ and its relatively low cost, sulfurization is a basic operation in winemaking; it was used by the ancient Romans in the era before Christ [2]. In wine, sulfur dioxide serves as a reducing agent; it has the ability to bind oxygen and other such elements in order to protect other species from oxidation, thereby keeping the wine fresh without oxidative browning. Its antimicrobial effects on various fungi and bacteria are also very important, alongside its ability to preserve the correct sensory properties of a given wine (such as its color, taste, and overall good health) [2]. Despite the positive effects of sulfur dioxide on wine, it is considered a xenobiotic substance that can have negative effects on the human body. Therefore, the EU has defined an amount of total sulfur dioxide in wine intended for consumption or sale that must not be exceeded. The daily dose of sulfur dioxide for a 70 kg person should not exceed 50 mg [3].

For these reasons, we need sensitive, highly selective, fast, and chemical-saving analytical detection techniques for the free and bonded sulfur dioxide in wine. However, as can be gleaned from the literature, few modern analytical methods have been developed. Except for classical volumetric methods, popular procedures involve instrumental techniques for the detection of sulfur dioxide. Isotachophoresis has been successfully applied for the detection of free and bonded SO_2_ in wine, with good analytical parameters (except the time required for analysis) [4]. The headspace of analytical tubes is also used for the determination of sulfur dioxide. This is a simple, fast, and reliable technique, but involves colorizing organic agents to produce a signal; moreover, this technique’s sensitivity is slightly lower [5]. Its sensitivity may be enhanced in the case of headspace gas chromatography [6] and headspace microextraction, followed by practical and specific SERS detection [7,8]. This method enables the analysis of many samples at a time, but it involves expensive instrumentation; this method’s working expenses are relatively high and are similar to those of nuclear magnetic resonance [9], wherein the high price of analysis is due to the extremely expensive instrumentation and the requirement of liquid helium for cooling. A very efficient, sensitive, and selective gas diffusion analytical system with indirect pH detection was introduced in [10,11,12]; however, this system also involves analyte separation through a membrane. Other popular methods involve FIA detection platforms based on the sequential analysis of many samples via potentiometric [13], chemiluminescent [14], and electrochemical [15] detection.

Voltammetry on a solid electrode combined with a chemical reaction in its diffusion layer was first introduced by Albery [16]. This detection platform uses a rotating ring disc electrode (RRDE) under bipotentiostatic conditions. Iodine from iodide is generated on the disc at a constant potential corresponding to the iodide oxidation. The ring is polarized by the potential for iodine reduction, and serves as an amperometric collector of unreacted iodine if a chemical reaction (e.g., with arsenic) takes place in the diffusion layer of the RRDE. However, the collection efficiency is only 40%, and has low current repeatability due to brush contacts. In addition, the species that react slowly with the electrogenerated species are difficult to detect due to the complicated shape of the generation–collection dependencies [17].

This approach was substantially improved by the introduction of an individually polarizable, comb-shaped pair of interdigitated microelectrodes [18]. These devices have extremely high collection efficiencies of around 90–100%, and are suitable for slow chemical reactions due to their overlapped diffusion layers; they also present the opportunity to lead the generation–collection process in the opposite direction through simple diffusion, instead of using less repeatable hydrodynamic transport, as in the case of RRDE [19,20]. In further research, it was shown that this process can also be accomplished by using only one working electrode polarized with a linear voltammetric scan. Under these conditions, iodide, as a product of a chemical reaction in the diffusion layer, diffuses back to the electrode surface and reoxidizes itself again, together with an incoming flux of iodide from the bulk phase solution. This substantially enhances the current response of the sensor [21,22]. The aforementioned approach to electrocatalytic recycling was successfully applied to the analysis of real samples, but its matrix was relatively simple. In this paper, this promising sensing platform is extended to samples of wine, which have a rather complex matrix. An unmodified boron-doped diamond electrode [23,24] was used as a working electrode because of its analytically advantageous features, such as an extremely low background current [25,26], a limited ability to adsorb species under normal conditions [27,28], and a wide range of working potentials [29,30].

## 2. Results and Discussion

First, we investigated the feasibility of direct determination of SO_2_ in wine. It is well known that wine samples have a rather complex matrix formed of various carboxylic acids, which may then behave as surfactants to block the electrocatalytic centers on the surface of a boron-doped diamond electrode. Therefore, potassium hexacyanoferrate in 0.1 mol dm^−3^ KCl was used as a supporting electrolyte, as in a typical testing solution, for the boron-doped diamond surface. In corresponding experiments, 0, 1, 2, and 3 mL of white wine were pipetted into volumetric flasks, and the volume was adjusted to 25 mL by adding 5 × 10^−4^ mol dm^−3^ of K_4_[Fe(CN)_6_] in KCl. Subsequently, these solutions were transferred into electrochemical cells, and cyclic voltammograms were performed in the potential range of −0.5 to +1.0 V vs. Ag/AgCl. These voltammograms are depicted in Figure 1A. As can be seen from this figure, the relative decrease in the anodic signal of K_4_[Fe(CN)_6_] was higher between 0 and 1 mL in comparison to 1 and 2 mL or 2 and 3 mL of added wine. The reason for this higher relative decrease in signal the presence of 1 mL wine (in comparison to 0 mL (except for dilution, diffusion-coefficient change, negligible matrix adsorption, and lower change in solution conductivity)) was the significant kinetics of charge-transfer mitigation, itself caused by pH change and ionic strength enhancement, which supports the association of the ethanol in wine with [Fe(CN_6_)]^−4^ anions. In the cases in which 2 or 3 mL of wine was present in the solution, the pH change was very small, and the change in the degree of association and ionic strength was also low in comparison with the solution with 1 mL. Therefore, the increments of the signal decrease resulting from the kinetics’ mitigation were lower, and the relative change in the signal was lower for 1 to 2 mL in comparison 0 to 1 mL of added wine [31]. The acidic medium was also very suitable for the anodic electrogeneration of halogens from halides, hence the signal at 800 mV is attributed to formation of Cl_2_ from KCl in this case.

In Figure 1B, it is shown that the voltammetric signal of the anodic oxidation of K_4_[Fe(CN)_6_] linearly (R^2^ = 0.99) diminished with its content in wine (Figure 1B). The relative decrease in the signal from 1 to 3 mL of wine was 1.28, and the relative decrease in the concentration of K_4_[Fe(CN_6_)] with the wine’s dilution was 1.10. This 16% relative deviation is attributed mainly to change in the diffusion coefficient caused by the increased concentration of species from wine.

To conclude, the voltammetric signal of the testing species in the presence of 1, 2, and 3 mL of wine remained immune to deforming, and was decreased mainly via simple dilution; moreover, negligible matrix adsorption took place during this process, which enabled such a direct analysis of the wine. Based on these results, 1 mL of wine was pipetted into 25 mL electrochemical cells in all the real-sample analyses.

The determination of sulfur dioxide in wine described herein is based on the anodic electrogeneration of iodine from iodide on a boron-doped diamond electrode, according to the following well-known equation:2I^−^ − 2e^−^ → I_2_(1)

As we observed in our previous paper [22], the charge transfer involved in the anodic oxidation of iodide to iodine is irreversible, but the oxidation peak recorded via cyclic or linear sweep voltammetry is sharp and very well developed; therefore, it is suitable for analytical purposes. When SO_2_ is added to the measured solution, the electrogenerated iodine reacts with the sulfur dioxide in the diffusion layer of the boron-doped diamond electrode, according to a relatively fast Bunsen reaction:SO_2_ + I_2_ + 2H_2_O → H_2_SO_4_ + 2I^−^ + 2H^+^(2)

As can be seen from Figure 2, the product of this chemical reaction, iodide, diffuses back to the electrode surface due to its concentration deficiency, and together with the incoming flux of iodide from the bulk phase of the solution, reoxidizes again to close the catalytic redox cycle; this enhances the voltammetric response of the sensor, even if an electroinactive species is added to the solution.

In the next stage of our research, it was necessary to optimize the experimental conditions to obtain the most effective analytical performance using this detection platform. The most important parameters are the pH of the supporting electrolyte and the initial concentration of KI. For studying pH, we used three various electrolytes of the same concentration of 0.1 mol dm^−3^, and the concentration of KI was 2 × 10^−5^ mol dm^−3^. A linear sweep voltammogram from 0.15 to 1.2 V vs. Ag/AgCl was registered with scan rate of 50 mV s^−1^; when Na_2_SO_3_ was added to the solution, a concentration of 4 × 10^−4^ mol dm^−3^ was registered. The peak currents were compared, and their difference was proportional to the signal amplification produced by the chemical redox cycling that took place in the diffusion layer of the boron-doped diamond electrode. As can be seen from Figure 3A, we obtained the highest signal amplification in 0.1 mol dm^−3^ HClO_4_, and practically no amplification was achieved when 0.1 mol dm^−3^ NaHCO_3_ with a pH value of 8.3 was used as the supporting electrolyte. In the case of the acetate buffer, we obtained significant enhancement of the voltammetric signal, but its magnitude was not as high as that obtained with perchloric acid. We also tried to use 0.05 mol dm^−3^ sulfuric acid, but found that the amplification was lower, because sulfate ions (as a product of the chemical reaction, and causing signal amplification) were in the solution in an excess proportion; therefore, the chemical redox cycling became kinetically impeded. For this reason, we used 0.1 mol dm^−3^ HClO_4_ as the supporting electrolyte in each further experiment. The corresponding linear sweep voltammograms of the anodic oxidation of KI in 0.1 mol dm^−3^ HClO_4_ in the absence and the presence of Na_2_SO_3_ are depicted in Figure 3B. No intrinsic signal of sulfur dioxide was observed in the absence of KI (dashed line in Figure 3B). From this figure, a potential shift in the presence of sulfite ions in the solution was also evident. This potential shift was proportional to the sulfite concentration, and may have been caused by the adsorption of sulfite on the BDD surface. We did not investigate this phenomenon further because it had no influence on the analytical characteristics (especially on the sensitivity of this determination). As for the initial concentration of KI, we chose 2 × 10^−5^ mol dm^−3^ as the optimal value, because at lower concentration, the amplification was very low; this was due to the very small regeneration diffusion flux of iodide, and also to the anodic peak of KI, which was not properly developed due to its very low concentration.

Under these optimized conditions, it was observed that the system was sensitive to addition of Na_2_SO_3_. In Figure 4A, linear sweep voltammograms are depicted for the 5 × 10^−5^ mol dm^−3^ solution of KI in 0.1 mol dm^−3^ HClO_4_, and six additions of 50 μL Na_2_SO_3_ with a concentration of 0.01 mol dm^−3^. The peak-current difference between certain addition of Na_2_SO_3_ and pure KI is ΔI_p_. This parameter represents signal amplification according to the mechanism depicted in Figure 2, and it was found that its dependence on the concentration of Na_2_SO_3_ was linear (Figure 4B), thereby fitting the equation ΔI_p_ =A + Bc with the following parameters: A = −1.11 × 10^−7^ A; s_A_ = 6.53 × 10^−8^ A; B = 0.02 A mol^−1^ dm^3^; s_B_ = 8.47 × 10^−4^ A mol^−1^ dm^3^; and R^2^ = 0.994. According to the 3s_A_/B criterion, the detection limit of 9 × 10^−6^ mol dm^−3^ was estimated. With further optimization, a detection limit of the order of 10^−7^ mol dm^−3^ could be reached [22]. This value of LOD is suitable for this kind of determination because the SO_2_ content in wine is at least 50 times higher, which allows suitable sample dilution and low relative standard deviations, thereby ensuring the high accuracy of this determination.

It is known that SO_2_ in wine is bonded by free aldehydes or the aldehydic parts of various molecules present in wine matrix. Chemical descriptions of this process are very scarce in the literature. If we take formaldehyde as the simplest example, in neutral or weakly acidic solution, sulfur dioxide is bonded by formaldehyde according to the following equation, which is relatively slow (it takes a few minutes):HCHO + H_2_SO_3_ → HCHOSO_2_ + H_2_O(3)

The structure of the compound HCHOSO_2_ is depicted in Figure 1.

Reaction (3) does not proceed in strongly alkaline and strongly acidic media, and compound HCHOSO_2_ does not react with electrogenerated iodine. When NaOH is added to the solution, the bonded SO_2_ is released to the solution freely, and may react with electrogenerated iodine. This process is described by the following equation:HCHOSO_2_ + 2 NaOH → HCHO + Na_2_SO_3_ + H_2_O(4)

After the addition of NaOH, it is necessary to wait a few minutes until all the SO_2_ is free; then, the pH can be altered to a strongly acidic state to obtain a good signal in the anodic oxidation of KI, a high rate of Bunsen reaction, and the stability of all the reactants. Bunsen reactions are not selective for SO_2_ from wine, and some species may react with electrogenerated iodine. For this reason, interferences should be considered. The necessary interference is relatively simple in this case. It is based on free SO_2_ bonding with formaldehyde, according to Equation (3), and on the addition of Chelaton III to make the analytes and metals present in wine inactive in the reaction with iodine. In addition, the signal of the anodic oxidation of iodide into iodine should be measured in the absence of any sulfur dioxide, and with the species reacting with iodine as a blank. It should then be subtracted from any signals measured, as demonstrated in Figure 5, wherein the selectivity of this determination is shown.

The sensing platform presented in this paper was validated using model samples with a known amount of Na_2_SO_3_, which was added to the supporting electrolyte. Because wine has inimitable features, it was difficult to prepare artificial samples that exactly simulated the matrix of wine; therefore, this validation served to verify the contents of free SO_2_. Three model samples were analyzed for their content of Na_2_SO_3_, which was of the order of 10^−5^ mol dm^3^. These results are summarized in Table 1. The given and found values were in statistical agreement, and the 95% confidence interval had relatively narrow halfwidth.
ijms-24-12875-t001_Table 1Table 1Model sample analysis of Na_2_SO_3_ (five parallel determinations).Na_2_SO_3_ Content (Given)10^5^ (mol dm^−3^)Na_2_SO_3_ Content (Found)10^5^ (mol dm^−3^)SD10^5^ (mol dm^−3^)^1^ RHW 95%(%) 11.10.054.132.90.123.754.80.183.4^1^ RHW: relative halfwidth of a 95% confidence interval; this halfwidth was calculated according to the following formula.
(5)$$\frac{t_{n-1,\alpha }SD}{\sqrt{n}}=\frac{2.132}{\sqrt{5}}SD$$

Finally, the sensing platform developed here was applied to a real analysis. Commercially available light white wines served as real samples.

In Figure 6, the method for calculating the amount of SO_2_ is presented. The curve b from Figure 5 represents blank and interfering species reacting with the iodine signal. This curve was subtracted from the LSV voltammograms depicted in Figure 6A, and values of ΔI_p_ were obtained. These values were used for the standard addition calibration plot in Figure 6B. The unknown concentration of SO_2_ in the diluted real sample was calculated as A/B of the aforementioned plot. The parameters of this plot are summarized in the caption of Figure 6B. This strategy is applicable only when using an equal amount of sample, and equal working volumes and concentrations; otherwise, a standard addition plot should be constructed for total, free SO_2_, and interferences.

Three real samples of white wine were analyzed, and the results of free SO_2_ content are presented in Table 2A; the results of total SO_2_ content are in Table 2B. 

As can be seen from these tables, the analyzed wines contained amounts of free SO_2_ that are typical of this kind of wine (Table 2A).

As for total SO_2_ (defined as free SO_2_ plus bonded SO_2_, released by NaOH treatment), in all samples, the 200 mg/L value defined by the EU was not exceeded. Our independent analytical technique served as a classical iodometric method for the determination of free and total SO_2_ contents. As can be seen from both tables, the results obtained by using both methods were in the statistical agreement, although the standard deviations were lower in the case of titrimetry, as observed in [4]. This is due to sample dilution, and to the analysis of substantially lower concentrations of SO_2_ in the case of the electrochemical method on a bare boron-doped diamond electrode. However, this method is many times faster, providing substantially more results for evaluation within a given time; moreover, the consumption of samples and other chemicals is many times lower, which may make this method environmentally cleaner compared with titrimetric methods. This method’s low price (in comparison with expensive optical methods), together with the acceptable electrode surface, means that this detection platform has real potential to became a routine analytical technique in the process of wine control.

## 3. Methods and Materials

All electrochemical experiments were performed using a universal modular electrochemical instrument, Autolab PGSTAT 302N (Metrohm Autolab BV, Utrecht, The Netherlands), which was equipped with NOVA 2.1.5 communication software. A three-electrode arrangement in a glass electrochemical cell with volume of 25 mL was used. The electrochemical cell was maintained at 25.0 ± 0.5 °C. A commercially available boron-doped diamond electrode (Windsor Scientific Ltd., Slough, UK) with a radius of 1.5 mm in a PEEK tube was used as the working electrode. As a reference electrode, a classical Ag/AgCl (3 mol dm^−3^ KCl) electrode was used, and a platinum macroelectrode with area of 1 cm^2^ was used as a counter. The surface of the working electrode was cleaned chronoamperometrically in 0.6 mol dm^−3^ H_2_SO_4_ at the constant potential of −3 V vs. Ag/AgCl for 300 s, and rinsed with triple-distilled water before each sequence of measurements. All measured solutions were purged with nitrogen for 5 min, because voltammetric response slightly diminishes with time, and signal stability worsens with time. This is due to the reaction of SO_2_ with dissolved oxygen. However, we checked the Na_2_SO_3_ stock solution every day, and it was found to be stable when stored on a laboratory table in the dark.

Deionized water (EUROWATER, Bratislava) was used for the preparation of all solutions. H_2_SO_4_ (0.1 mol dm^−3^, MIKROCHEM, Pezinok, Slovakia), HClO_4_ (0.1 mol dm^−3^, LACHEMA, Brno, Czech Republic), ABS solution (pH = 4.6; 0.2 mol dm^−3^ CH_3_COONa mixed with 0.2 mol dm^−3^ CH_3_COOH, LACHEMA, Brno, Czech Republic), KCl (0.1 mol L^−1^, LACHEMA, Brno, Czech Republic), and NaHCO_3_ with a pH of 8.3 (0.1 mol L^−1^, LACHEMA, Brno, Czech Republic) served as supporting electrolytes. Formaldehyde (37%, SKLOCHEM, Bratislava, Slovakia) was used in interference interrogation experiments. Similarly, all of the solid chemicals (NaOH (CENTRALCHEM, Bratislava, Slovakia), KI (Lachema, Brno, Czech Republic), Na_2_SO_3_ (LACHEMA, Brno, Czech Republic), K_4_[Fe(CN)_6_]·3H_2_O (CENTRALCHEM, Bratislava, Slovakia) and EDTA (LACHEMA, Brno, Czech Republic)) were of p.a. purity. A series of the solutions with various concentrations was prepared via dilution with stock solutions. The real samples were analyzed directly. We pipetted 1 mL of white wine and a suitable volume of the other chemicals (with a certain concentration) into an electrochemical cell in order to yield a desirable concentration with an adjusted final volume of 25 mL. A multiple standard addition calibration plot method was used for SO_2_ quantification, with known additions of Na_2_SO_3_.

## 4. Conclusions

A bare boron-doped diamond electrode was used in wine analysis for the first time. The detection platform presented here is based on the electrogeneration of iodine from iodide in anodic conditions. The electrogenerated iodine reacts further with SO_2_, according to Bunsen’s reaction, thereby forming electroinactive sulfate and an iodide anion that diffuses back to the electrode surface and reoxidizes again. This catalytic redox cycling enhances the voltammetric response of the sensor; even SO_2_ is electroinactive in this case. To achieve this effect with maximal current enhancement, the pH value of the supporting electrolyte and the concentration of KI should be optimized. The proposed methodology is suitable for the determination of the free (direct analysis) and total content of SO_2_ (i.e., NaOH treatment during sample preparation) in white wines, and also for the determination of interference species that react with electrogenerated iodine (i.e., via formaldehyde treatment during sample preparation). After full validation (except for interlaboratory measurements), this method was applied to real samples, which provided statistically similar results to those of classic titrimetry (an independent and widely used method for this kind of determination). A relatively low detection limit of 10^−6^–10^−7^ mol dm^−3^ allowed for dilution of a small amount of the sample and very low consumption of chemicals. Together with the low price, speed, and ability of this analysis to obtain substantially more results in real time, we may consider the methodology proposed herein as a cleaner alternative among techniques for wine analysis, with great potential for use as a routine method for this type of analysis.

## Data Availability

Not applicable.

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
