# Peer review of "An Interference-Free Voltammetric Method for the Detection of Sulfur Dioxide in Wine Based on a Boron-Doped Diamond Electrode and Reaction Electrochemistry"

_ijms, 2023, doi:10.3390/ijms241612875_

Round 1
Reviewer 1 Report
Reviewer’s recommendations, questions as well as inaccuracies found are listed below.
line 76 – lower
line 120 – hexacyanoferrate
line 127 – dilution
line 162 - is proportional to
lines 196-197 – should be presented as equation
line 212 – and
The stability of electrode surface in water solution after electrochemical pretreatment at -3V begs the questions. Authors concluded that the simple dilution takes place as the reason of hexacyanoferrate’s current decreasing. But even for 3 ml wine aliquot the dilution will be equal to 1.12 ((25+3)/25) regarding 0 ml aliquot and 1.08 regarding 1 ml aliquot. It causes the concentration decreasing by 1.08 times. According to Randles–Sevcik equation, this concentration decreasing will cause current decreasing by 1.08 times. Voltammograms presented in Figure 1 reveal current decreasing by approximately 1.56 times (from 2.8 mkA to 1.6 mkA) between 1 and 3 ml of wine. Moreover, peak-to-peak separation increases indicating the diminution of kinetics. For better understanding of processes occurred blank experiment should be performed with water or KCl solution instead of wine. Voltammograms obtained before dilution should be added as well as background voltammograms (in absence of hexacyanoferrate). Anodic signals at about 800 mV should be explained. Furthermore, the comparison of Fig.1A with Fig.1B gives rise to doubts: anodic peak current for 3ml aliquot looks obviously higher than 2 mkA, meanwhile it is about 1.6 mkA in Fig.1B.
Since sodium sulfite is well known oxygen removing agent the term “actual concentration” is not appropriate. Did authors use the oxygen removing procedure (purging with argon or nitrogen) before experiments? Voltammogram for sodium sulfite in absence of KI should be added to Figure 3B to evaluate the sulfite intrinsic signal.
The peak of KI in absence of sulfite has different position in Fig.3B and 4A, why so? KI concentration should be rechecked (50 mkM in Fig.4 caption).
The wine samples pretreatment should be revealed more clearly in experimental part. The necessity of NaOH addition followed by mixing with strong acidic solution (0.1M HClO4) should be explained and accompanied by corresponding reaction since authors claim the interaction of free sulfur dioxide (equation 2, Figure 2). The principle of sensor’s functioning should be discussed more clearly.
line 76 – lower
line 120 – hexacyanoferrate
line 127 – dilution
line 162 - is proportional to
lines 196-197 – should be presented as equation
line 212 – and
Author Response
Dear reviewer,
I am sending to you in the form of itemized coverletter response to your comments in the attached file

Reviewer 2 Report
The paper presents a novel analytical technique for detecting sulfur dioxide in wine, even in the presence of a complex matrix. The method utilizes a boron-doped diamond electrode and the electrogeneration of iodine from iodide, without any modifications.
Question:
-
Could you provide more information about how the selectivity of the determination was achieved using NaOH and formaldehyde during sample preparation?
-
In terms of practical application, how feasible is the use of a boron-doped diamond electrode for routine detection of sulfur dioxide in wine?
-
Does the paper discuss any potential limitations or challenges associated with the proposed analytical technique?
-
Are there any comparisons made between this new technique and existing methods for sulfur dioxide detection in wine? If so, what are the advantages or disadvantages of the proposed technique compared to others?
Author Response
Dear referee,
I am sending to you our response on your comments in the form of itemized coverletter in the attacched file

Round 2
Reviewer 1 Report
Revised version of manuscript looks more understandable and logical, but the following drawbacks should be revised before publication. I have no doubts that authors can improve the manuscript.
1. I still have doubts about the stability of electrode surface in aqueous solution after pretreatment and about the wine’s matrix effect. The same doubts would appear during reading by potential readers. Authors did not performed blank experiments (KCl solution instead of wine) mentioned in first revision. The increase of peak-to-peak separation as well as oxidation peak shift could not be caused by pH change since the hydrogen ions are not involved in electrochemical reaction of hexacyanoferrates. Even if it was incorrect the reduction peak would shift in the same direction with oxidation peak. “The main purpose of this study was to prove that signal decrease is caused mainly by simple dilution” – the simple replacement of wine with KCl solution is the best way to evaluate the dilution impact. My recommendation – Authors should either provide blank experiment or discuss it in text.
2. line 132 - eleCtrogeneration
3. line 209-210 – KI concentration is 20 mkM, at the same time it is 50 mkM in Fig.4 caption.
4. It seems that “strongly acidic media” (line 238) is not appropriate to sulfur dioxide “activation” since the addition of formaldehyde and wine to 0.1M HClO4 did not cause any increasing of current (Fig.5 curve b) in contrast with NaOH. The necessity of Chelaton III should be explained in text.
5. line 241 – process
6. According to text (lines 281-282) the determination of sulfur dioxide concentration in wine is accompanied by interference experiment, calibration curve plotting and A/B calculation. It is unclear from text: do we need to plot the curve for each wine sample?
7. The explanation for “total content” term is given only in Conclusions. It should be added in text where the Table 2B is discussed.
Round 3
Reviewer 1 Report
The manuscript has been improved, reviewer has received the detailed explanation of some unclear moments.